# Global Profiling of the Antioxidant Constituents in Chebulae Fructus Based on an Integrative Strategy of UHPLC/IM-QTOF-MS, MS/MS Molecular Networking, and Spectrum-Effect Correlation

**DOI:** 10.3390/antiox12122093

**Published:** 2023-12-08

**Authors:** Xiangdong Wang, Jian Xu, Li-Hua Zhang, Wenzhi Yang, Huijuan Yu, Min Zhang, Yuefei Wang, Hong-Hua Wu

**Affiliations:** 1State Key Laboratory of Component-Based Chinese Medicine, National Key Laboratory of Chinese Medicine Modernization, Institute of Traditional Chinese Medicine, Tianjin University of Traditional Chinese Medicine, 10 Poyanghu Road, West Area, Tuanbo New Town, Jinghai District, Tianjin 301617, China; xiangdongblue@163.com (X.W.); xjwyq123456@163.com (J.X.); zhanglihua200061@163.com (L.-H.Z.); wzyang0504@tjutcm.edu.cn (W.Y.); huijuanyu@tjutcm.edu.cn (H.Y.); 2Haihe Laboratory of Modern Chinese Medicine, 10 Poyanghu Road, West Area, Tuanbo New Town, Jinghai District, Tianjin 301617, China

**Keywords:** *Terminalia chebula*, Chebulae Fructus, tannins, molecular networking, spectrum-effect relationship

## Abstract

An integrative strategy of UHPLC/IM-QTOF-MS analysis, MS/MS molecular networking (MN), in-house library search, and a collision cross-section (CCS) simulation and comparison was developed for the rapid characterization of the chemical constituents in Chebulae Fructus (CF). A total of 122 Constituents were identified, and most were phenolcarboxylic and tannic compounds. Subsequently, 1,3,6-tri-*O*-galloyl-*β*-d-glucose, terflavin A, 1,2,6-tri-*O*-galloyl-*β*-d-glucose, punicalagin B, chebulinic acid, chebulagic acid, 1,2,3,4,6-penta-*O*-galloyl-*β*-d-glucose, and chebulic acid, among the 23 common constituents of CF, were screened out by UPLC-PDA fingerprinting and multivariate statistical analyses (HCA, PCA, and OPLS-DA). Then, Pearson’s correlation analysis and a grey relational analysis were performed for the spectrum-effect correlation between the UPLC fingerprints and the antioxidant capacity of CF, which was finally validated by an UPLC-DPPH^•^ analysis for the main antioxidant constituents. Our study provides a global identification of CF constituents and contributes to the quality control and development of functional foods and preparations dedicated to CF.

## 1. Introduction

*Terminalia chebula* Retz. (Combretaceae) is a *Terminalia* tree native to South and Southeast Asia [1]. Its dried fruit, namely Chebulae Fructus (CF), is a well-known, common folk medicine widely used in Tibetan and Ayurvedic systems of medicine [2] in countries such as China, India, and Nepal [3]. CF is recognized as “the king of medicine” and frequently prescribed in traditional medicinal prescriptions with valuable and diverse pharmacological potentials, such as antioxidant [4], anti-inflammatory [5], anti-diabetic [6], antimicrobial [7], and virustatic [8] activities. Modern pharmacological studies have found that CF exerted predominantly antioxidant activity by relying on its high biological yield of phenolcarboxylic and tannic antioxidants [9], as exemplified by gallic acid, ellagic acid, chebulic acid, corilagin, chebulain, and chebulagic acid, with great intrinsic potential for preventing oxidative stress and inhibiting reactions caused by oxygen or peroxides, as well as counteracting the destructive effects of oxidation in animal tissues in order to reap further anti-aging benefits [10] for the body. Among the edible fruits in the Himalayan region of India, CFs with the highest antioxidant level have been used as a daily dietary supplement among people living in the mountains [11]. A recent study revealed that corilagin, chebulagic acid, and chebulinic acid in CF may be absorbed into the body for a comparatively long term to exert continuous treatment effects [12].

In previous studies, a cellulose filter paper (CFP)-immobilized AChE assay, coupled with UPLC-(-)-ESI-QTOF-MS and molecular docking, was developed to screen and identify 26 potential AChE inhibitors, including corilagin, 1,3,4-tri-*O*-galloyl-*β*-d-glucose, chebulagic acid, ellagic acid, and chebulinic acid, from the fruits of *T. chebula* [13]. And 14 main constituents (gallic acid, chebulic acid, 1,6-di-*O*-galloyl-d-glucose, punicalagin, 3,4,6-tri-*O*-galloyl-d-glucose, casuarinin, chebulanin, corilagin, neochebulinic acid, terchebulin, ellagic acid, chebulagic acid, chebulinic acid, and 1,2,3,4,6-penta-*O*-galloyl-d-glucose) from the bark and/or the fruits of *Terminalia* plants were determined by using the HPLC method [14]. Overall, the relatively diverse and complex structure, as well as the wide molecular weight distribution of tannins, may hinder the quality evaluation of CF [15], resulting in a lack of acknowledged quality standards for CF, at present. Therefore, sensitive, specific, and precise analytical methods are still imperative for the characterization of the constituents and the standardization of CF.

As is known, the difficulty in researching CF lies in the complexity of components and multi-bioactivities, whose associations have not been systematically unveiled, making it difficult to systematically elucidate CF’s effectiveness [16]. Mass spectrometry (MS)-based, non-targeted metabolomics have become an effective method for the identification of bioactive constituents in Chinese medicines (CMs) because of the high throughput, fast screening, and wide coverage [16]. The Global Natural Products Social (GNPS)-based molecular networking (MN) is a computational method that bridges the gap between popular mass spectrometric (LC-MS/MS) data-processing and molecular networking analysis, by calculating the similarities between the MS/MS spectral pairs in GNPS [17]. Recently, LC-MS/MS, coupled with molecular networking, has enabled the identification of the main constituents, such as ellagic acid and its derivatives, from *T. leiocarpa* (DC.) Baill [18]. And spectrum-effect relationship modeling; enzymatic- and chemical-reagent-based online chromatographic screening [19]; and biological target fishing [20] have been developed as optional methods for the fast discovery and screening of active constituents from CMs.

In this study, an integrative strategy of GNPS-based MS/MS classic molecular networking, an in-house library search, and a simulation and comparison of collision cross-sections (CCS) was developed firstly based upon the UPLC/IM-QTOF-MS analysis for the rapid identification of the phenolcarboxylic and tannic metabolites in CF (Figure 1). Next, the antioxidant capacity of CF was evaluated by DPPH^•^-scavenging, ABTS^+^-scavenging, and ferric-reducing assays, and then Pearson’s correlation analysis and a grey relational analysis were performed to assess the spectrum–antioxidant relationship between UPLC fingerprints and the antioxidant effects of CF. Finally, a comparative UPLC analysis, before and after treatment of DPPH^•^, was further applied to verify the main antioxidants in CF.

## 2. Materials and Methods

### 2.1. Chemicals and Reagents

Methanol (chromatographically pure, Sigma-Aldrich Trading Co., Ltd., Darmstadt, Germany), anhydrous ethanol (analytically pure, Tianjin Concord Technology Co., Ltd., Tianjin, China), formic acid (chromatographically pure, Shanghai Aladdin Biochemical Technology Co., Ltd., Shanghai, China), and purified water (Guangzhou Watsons Food and Beverage Co., Ltd., Guangzhou, China) were used in this study. Gallic acid (≥98%, B20851), punicalagin (≥98%, B20013), 1,3,6-tri-*O*-galloyl-*β*-d-glucose (≥98%, B50707), ellagic acid (≥98%, B21073), and DPPH^•^ (≥98.0%, B25609) were purchased from Shanghai Yuanye Bio-Technology Co., Ltd., Shanghai, China. DIAION HP-20 macroporous resin was obtained from Mitsubishi Chemical Corporation, Tokyo, Japan. Methanol-*d*_4_ (99.8%) and dimethyl sulfoxide-*d*_6_ (DMSO-*d*_6_, 99.9%) were obtained from Cambridge Isotope Laboratories, Inc., Andover, MA, USA.

### 2.2. Plant Material

Eighteen batches of CF (whole fruits or the flesh of fruits) were collected from different countries and regions in 2021, with detailed information, as shown in Appendix A. All the collected materials were authenticated as the dried ripe fruits of *Terminalia chebula* Retz. (Samples **S1**–**S18**) based on the DNA-barcoding and PCR method, as reported by Prof. Xiaoxuan Tian from the Institute of Traditional Chinese Medicine at the Tianjin University of Traditional Chinese Medicine. As recorded in the Chinese pharmacopoeia 2020 Edition [21], the dried, ripe fruits of *T. chebula* were collected, and their impurities were cleaned from the surface before being washed and dried in the sun, resulting in CF fruits. CF fruits were cleaned and briefly soaked to moisten them enough to remove the kernels, before being dried in the sun to yield the flesh of CF fruits. The dried materials were pulverized by a shredder BJ-800A (Hangzhou Baijie Technology Co., Ltd., Hangzhou, China) and filtered through a 50-mesh sieve to yield CF powder, before use.

### 2.3. Extraction, Isolation, and Identification of the Main Constituents in CF

The crushed, ripe fruits of *T. chebula* Retz. (5.0 kg, Hebei Chunkai Pharmaceutical Co., Ltd., Baoding, Hebei, China) were firstly extracted by soaking in 20 L 50% aqueous ethanol at room temperature for 48 h and then ultrasonically extracted (1000 W, Kunshan Ultrasonic Instrument Co., Ltd., Kunshan, Jiangsu, China) for 2 h. The extractions were combined and concentrated to dryness at 40 °C by rotary evaporator (Rotavapor R-250, BUCHI, Flawil, Switzerland) to yield a crude extract (1.60 kg). The crude extract (1.50 kg) was then dissolved in 5 L water and subjected to a column chromatography (50 cm × 10 cm) of DIAION HP-20 macroporous resin, which was then eluted successively by water and 95% ethanol to afford fractions F1 (900 g) and F2 (390 g), respectively. The major constituents were isolated from the crude extract (15.0 mg/mL) by preparative HPLC (1260 Infinity II, Agilent, Santa Clara, CA, USA) and identified as chebulic acid (*t*_R_ = 22.1 min) [4], gallic acid (*t*_R_ = 23.5 min) [22], 1-*O*-galloyl-*β*-d-glucose (*t*_R_ = 27.3 min) [22], 4-*O*-galloyl-(-)-shikimic acid (*t*_R_ = 42.3 min) [23], 5-*O*-galloyl-(-)-shikimic acid (*t*_R_ = 50.1 min) [23], corilagin (*t*_R_ = 59.8 min) [22], 1,3,6-tri-*O*-galloyl-*β*-d-glucose (*t*_R_ = 67.3 min) [24], urolithin M5 (*t*_R_ = 73.1 min) [4], chebulanin (*t*_R_ = 74.8 min) [4], chebulagic acid (*t*_R_ = 86.4 min) [4], chebulinic acid (*t*_R_ = 93.2 min) [4], ellagic acid (*t*_R_ = 96.3 min) [22], eschweilenol C (*t*_R_ = 97.5 min) [4], terminalin (*t*_R_ = 100.1 min) [25], and 4-*O*-(3″,4″-di-*O*-galloyl-*α*-l-rhamnosyl)ellagic acid (*t*_R_ = 105.1 min) [4], respectively, by comparison of their NMR (^1^H/^13^C, 600 MHz/150 MHz, Bruker, Zurich, Switzerland) data (Appendix A) with those reported. They were isolated by a preparative HPLC with a gradient elution of methanol (A) and 0.1% formic acid water (B) from 5% to 95% (A) in 90 min through a COSMOSIL PBr column (20 mm × 250 mm, 5 μm), which was then rinsed for 20 min, after one preparation. These isolated compounds were all above 95% in purities of HPLC-grade and were kept in the dark at 4 °C, before use.

### 2.4. UHPLC-QTOF-MS and UPLC/IM-QTOF-MS Analyses

CF crude extract, fractions F1 and F2, and the mixed standards solution (Appendix A) were analyzed by an ultra-high performance liquid chromatographic system (Agilent 1260 Infinity II) coupled with an Agilent 6550 QTOF™ high-resolution mass spectrometer (Agilent, CA, USA). Chromatographic separation was achieved on a COSMOSIL PBr column (2.1 mm × 100 mm, 2.6 μm) at 30 °C. The mobile phase of 0.1% formic acid in water (A) and methanol (B) ran at a flow rate of 0.3 mL/min. The injection volume was 5 μL and pre-equilibrated for 10 min. The elution program was undertaken according to the following gradient: 0–5 min, 0–5% B; 5–9 min, 5–30% B; 9–12 min, 30–33% B; 12–19 min, 33–50% B; 19–28 min, 50–81% B; 28–30 min, 81–100% B; 30–33 min, 100% B; 33–36 min, 100–0% B; and 36–40 min, 0% B. All samples were injected once in a negative ion mode. Sample preparation and the QTOF-MS source parameters are detailed in the Appendix A. UHPLC-QTOF-MS analysis.

Based on the aforementioned chromatographic separation, UPLC/IM-QTOF-MS analysis and CCS measurement were performed on a Waters UPLC/VION IMS-QTOF-MS system (Milford, MA, USA), following a reported identification method [26], as detailed in the Appendix A. UPLC/IM-QTOF-MS analysis. The CCS prediction was accomplished by AllCCS (http://allccs.zhulab.cn/, last accessed on 10 October 2023).

### 2.5. GNPS-based MS/MS Classic Molecular Networking

Molecular networking is a spectral correlation and visualization approach that visually displays the chemical space present in tandem mass spectrometry (MS/MS) experiments. GNPS was used for MS/MS molecular networking that visualized sets of MS/MS spectra among related molecules (called spectral networks), even when the MS/MS spectra themselves had not been matched to any known compounds [17]. In MS/MS molecular networks, each MS/MS spectrum was displayed as a node, and the spectrum-to-spectrum components were aligned as an edge, and the authors observed that similar spectra from structurally related molecules tended to be clustered in one component when these molecular fragments were similar, as reflected in their MS/MS patterns [27].

Classical molecular networking is well suited for discovery and can be analyzed directly from raw mass spectrometry files. Before GNPS-based MS/MS classic molecular networking, all MS data files were converted from the .d file format (Agilent MassHunter) to the .MzXML format by the software of MSConvert 3.023083, a standard tool for significant inquiries [28]. Through an FTP client, such as the software of WinSCP 5.17.9, the .mzXML documents were upload to GNPS (https://gnps.ucsd.edu/, last accessed on 10 November 2022) for further data analysis. In this study, the mass spectrometry files of fractions F1 and F2, the only two fractions derived from the CF total extract, were uploaded. The spectrum files G1, G2, and G6 were F1, F2, and blank, respectively, and G6 as the blank spectra was filtered before networking (https://gnps.ucsd.edu/ProteoSAFe/status.jsp?task=8daa34c6c40849109f4c08302b0f2e01/, last accessed on 12 November 2022). The tolerances of the precursor and the fragment ion mass were set at 0.01 Da, and all other parameters were defaults.

In addition, the MolNetEnhancer reanalysis of the molecular networking was conducted for automatic chemical classification through ClassyFire [29] to provide a more comprehensive chemical overview while illuminating structural details for each fragmented spectrum (https://gnps.ucsd.edu/ProteoSAFe/status.jsp?task=b66535212e0c4027b88a0a2a96c75324/, last accessed on 12 November 2022). Finally, the cytoscape file was downloaded and visualized by Cytoscape 3.8.2 software for further network analysis.

### 2.6. Establishment of In-House Compound Library

An in-house compound library of currently reported constituents of CF was established by a comprehensive literature review of online databases, including SciFinder^®^, ScienceDirect (https://www.sciencedirect.com/, last accessed on 1 November 2022), PubMed (https://pubmed.ncbi.nlm.nih.gov/, last accessed on 1 November 2022), and CNKI (https://www.cnki.net/, last accessed on 10 October 2022), combined with the HMDB database (https://hmdb.ca/, last accessed on 15 November 2022). The comprehensive molecular information (such as the 2D and 3D structures, molecular formula, molecular weight, exact mass, and the reported MS pattern) of more than 140 compounds (including phenolcarboxylic and tannic compounds, flavonoids, and triterpenoids) were included and collated (Appendix A). By applying a built-in library-search utility using the Agilent MassHunter workstation 10.0 software, the constituents in CF were preliminarily identified by molecular formula, as well as by MS or MS/MS library search.

### 2.7. UPLC Fingerprinting and Multivariate Statistical Analysis

Each CF powder (0.2 g) was accurately weighed, placed into a 50 mL volumetric flask, and ultrasonically extracted (600 W, Zhixin Instrument Co., Ltd., Shanghai, China) with 50 mL methanol, for 20 min at 30 °C. The extraction of each CF sample was then centrifuged (17,709× *g* for 10 min) at room temperature before direct injection, for UPLC fingerprinting.

The UPLC fingerprinting method was developed on a Waters H-Class plus UPLC-PDA system (Milford, MA, USA) equipped with a COSMOSIL PBr column (2.1 mm × 100 mm, 2.6 μm), at 30 °C. The mobile phase was composed of 0.1% formic acid in water (A) and methanol (B) at a flow rate of 0.3 mL/min, and the gradient elution program was set at 0–5 min, 0–5% B; 5–9 min, 5–30% B; 9–12 min, 30–33% B; 12–19 min, 33–50% B; 19–28 min, 50–81% B; 28–30 min, 81–100% B; and 30–33 min, 100% B. The injection volume was 2 μL. The ultraviolet absorption spectra of all the samples were recorded in the range of 190–400 nm, and the detection wavelengths were set at 254 nm and 270 nm.

After UPLC analysis, the original data files with the suffix “.cdf” were introduced into the Similarity Evaluation System for Chromatographic Fingerprint of TCM (Version 2012A) software, for the generation of the chromatographic fingerprints, the assignment of the common peaks, and the calculation of spectral similarities. Multivariate statistical analyses, including HCA, PCA, and OPLS-DA, were conducted on the chromatographic data of 18 batches of CF samples for non-targeted sample classification and the screening of differential metabolites by the R4.2.2-programming language and the SIMCA-P (version 14.1) software (Umetrics, Umeå, Sweden). Heatmaps were produced using the heatmap package in the R4.2.2-programming language. 

### 2.8. Antioxidant Assays, Spectrum-Effect Correlation, and DPPH^•^ Pretreated UPLC Analysis

The antioxidant capacity of CF was evaluated by DPPH^•^ and ABTS^+^ radical-scavenging and ferric-reducing antioxidant-power (FRAP) assays, following previously reported methods [30,31,32], with minor changes as detailed in the Appendix A. Evaluation of antioxidant capacity. The spectrum–antioxidant relationship of CF was then studied by Pearson’s correlation analysis [33] and grey relational analysis [34], between relative contents (%) of the selected common constituents and the antioxidant effect (IC_50_ values of the DPPH^•^ radical-scavenging effect, the ABTS^+^ radical-scavenging capacity at a concentration of 0.05 mg/mL, and the ferric-reducing antioxidant power at a concentration of 0.5 mg/mL) of the CF methanol extracts. A comparative UPLC-PDA analysis was performed, before and after treatment with DPPH^•^ (viz. UPLC-DPPH^•^ analysis) [31], for a selected CF sample (sample **S5**). The tested solution of CF methanol extract (CFME, 1.0 mg/mL) was mixed with an equal volume of DPPH^•^ (1.0 mmol/L) solution and incubated in the dark at room temperature for 30 min before being centrifuged and injected for UPLC analysis by applying the above UPLC fingerprinting method, with a solution of CF methanol extract (CFME, 1.0 mg/mL), mixed with an equal volume of methanol, as the control sample (*n* = 6).

## 3. Results and Discussion

### 3.1. Integrative Strategy for Global Identification of the Constituents in CF

Herein, an integrative strategy combining a UPLC/IM-QTOF-MS analysis, a GNPS-based MS/MS classic molecular networking, an in-house compound library search, and a CCS simulation and comparison was firstly developed for the global characterization of the chemical constituents in CF (Figure 1).

Firstly, the CF crude extract and the derived fractions (F1 and F2) were subjected to a UHPLC-DAD-(-)-QTOF-MS analysis before the GNPS-based MS/MS classic molecular networking, where the network nodes were automatically annotated by a spectral library search with the corresponding MS/MS spectra. Then, the molecular network was further re-analyzed by a MolNetEnhancer utility to yield enhanced results (Figure 2A) by integrating metabolomic mining and annotation tools [29]. Secondly, we established an in-house compound library that consisted of a compound name, a molecular formula, and a molecular weight, of each constituent of CF as reported, combined with the HMDB database for matching the observed compounds’ MS/MS spectral data. This step could be used to ensure the comprehensiveness and the novelty of the identification results because most of the constituents in the literature have not yet been included in the existing online databases. Thirdly, the MS cleavage law of specific compounds was summarized from the fragmentation patterns of the standard constituents, so the unknown constituents with similar fragmentation patterns could be inferred accordingly with the molecular formula, the glycosyl type, and the sequence of phenolcarboxylic or tannic substituents. This could be a key method for the discovery of novel compounds. The results of the molecular networking, the library search, and the fragmentation reference of the standard constituents were then combined and de-duplicated for a more global identification of the constituents in CF. Finally, the UPLC/IM-QTOF-MS analysis and the CCS simulation and comparison (Table 1) were used to assign more accurate chemical structures for the identified isomers. Meanwhile, a spider-web diagram was drawn based on the total-ion-current chromatographic (TIC) peak areas of the identified phenolcarboxylic and tannic constituents in CF (Figure 2B).

#### 3.1.1. Identification Based on Molecular Networking and Annotation of the Network Nodes

The GNPS-based MS/MS classic molecular network of the CF crude extract and the derived fractions F1 and F2 was constructed, as shown in Figure 2A. In the molecular network, each node contained the information of the precursor mass, retention time, and the MS/MS fragmentation, and the nodes with highly similar MS/MS patterns were clustered and classified into one network. There were 430 nodes in total, with the exception of the orphaned nodes due to their inability to reflect the relationships among the MS/MS spectra. However, due to the limited number of shared compounds collated in GNPS-cooperative online public spectral libraries, there were only 19 nodes auto-annotated in CF’s molecular network (https://gnps.ucsd.edu/ProteoSAFe/result.jsp?task=8daa34c6c40849109f4c08302b0f2e01&view=view_all_annotations_DB/, last accessed on 12 November 2022). The molecular network was then re-analyzed by MolNetEnhancer to yield 4 tannic components (6, 17, 13, and 14) with 132 nodes, 1 triterpenoid component (4) with 12 nodes, 2 flavonoid components (3 and 30) with 4 nodes, and other components with less nodes (Figure 2A). Based on the quasi-molecular ions, the fragment ions, and the adduct ions of the annotated standard compounds and the MS/MS spectral similarities among the nodes, the chemical structures of other known and/or unknown compounds (i.e., MN-01–MN-37, Appendix A) and the derivative pathways were extrapolated, as shown in Appendix A. In other words, our study initially relied on classic molecular networking to pursue unknown compounds based on the known compounds of CF extract, and the derived fractions based on their MS/MS fragmentation patterns.

In the molecular network, the nodes of gallotannins had accumulated predominantly in component 6**,** including monomeric, dimeric, trimeric, tetrameric, and polymeric galloyl tannins. Upon comparison with the MS/MS characteristics of the known 4-*O*-galloyl-(-)-shikimic acid and 5-*O*-galloyl-(-)-shikimic acid (*m*/*z* 325.057, [M–H]^−^), we could speculate about the structures of 3-*O*-galloyl-(-)-shikimic acid (*m*/*z* 325.057, [M–H]^−^) as an isomer, di-*O*-galloyl-(-)-shikimic acid (*m*/*z* 477.067, [M–H]^−^) as a dimer, and tri-*O*-galloyl-(-)-shikimic acid (*m*/*z* 629.079, [M–H]^−^) as a trimer. However, whether the result was monogalloyl-*β*-d-glucose (*m*/*z* 331.0669 [M–H]^−^) or polygalloyl-*β*-d-glucose could not be convincingly annotated because a large number of isomers had been generated by the presence of multiple acylation sites on the glycose or via polyol moiety.

Components 17 and 13 included mainly the chebulic ellagitannins featuring chebuloyl scaffolds in multiple derivative forms. For example, the two carboxyl groups in a chebulic acid could be dehydrated to form an anhydrate fragment ion (*m*/*z* 337.0201), while other types of chebuloyl (Che) in chebulanin, chebulagic acid, and chebulinic acid exposed three carboxyl groups by opening of the glycoside ester bond to form fragment ions at *m*/*z* 337.0201. In component 17, there were two kinds of chebulic ellagitannins in CF: One was simple chebulic acid polyol esters, and the other was tannins containing the Che unit. As compared to the quasi-molecular ion of chebulic acid (*m*/*z* 355.031, [M–H]^−^), the mass differences of the nodes at *m*/*z* 369.047, 383.062, and 411.093 were 14.016, 28.031, and 56.062 Da, respectively, indicating that they were mono-methylated, di-methylated, and tetra-methylated products of chebulic acid.

Component 13 contained chebulic ellagitannins, also called chebulanin derivatives. The node at *m*/*z* 1303.18 was the [2M–H]^−^ ion of chebulanin. As compared to the quasi-molecular ion of chebulanin (*m*/*z* 651.0839 [M–H]^−^), the structures of MN-29, MN-30, MN-31, and MN-33 were deduced accordingly from the nodes at *m*/*z* 815.152, 813.136, 807.126, and 803.094, respectively, with mass differences of 164, 162, 156, and 152 Da, respectively. The 1′-*O*-methyl neochebulanin node with a precursor mass at *m*/*z* 683.110 was deduced by following our in-house compound library search; thus, the dimethyl neochebulanin node at *m*/*z* 697.126 with a mass difference of 14 Da (as compared to *m*/*z* 683.110) could be deduced accordingly, which had been derived by a cleavage of 2-glycosidic bond in chebulanin, followed by a mono-methylation and a di-methylation of the exposed carboxyl groups.

The ellagitannin nodes were typically clustered in component 14 for the fragmentation of a common fragment ion at *m*/*z* 300.9998. The ellagitannins could be identified accordingly and were distinguished based on their characteristic hexahydroxydiphenoyl (HHDP) fragments (Figure 2C), such as MN-12 (*t*_R_ = 13.86 min, *m*/*z* 783.0672) and terflavin B (*t*_R_ = 15.80 min, *m*/*z* 783.0679). The predominant fragment ion of MN-12 was *m*/*z* 481.0663, which was a characteristic fragment ion of HHDP-glucose, while the predominant fragment ion of terflavin B was *m*/*z* 450.9940, which was a characteristic fragment ion of flavogallonic acid. In addition, component 14 also contained nodes of ellagitannins with rhamnose substituted by ellagic acyls, which could have been generated by the hydrolysis of HHDP, such as **116** (MN-15) and **118** (4-*O*-(4″-*O*-galloyl-*α*-l-rhamnosyl)ellagic acid) in Table 1.

Notably, the MS profiles of gallic acid, chebulic acid, chebulanin, and ellagic acid played crucial roles in the annotation of the nodes in components 6, 17, 13, and 14 (Appendix A), respectively.

#### 3.1.2. Identification by in-House Library Search

Forty-seven compounds were preliminarily identified by an in-house library search based on the auto-MS/MS mode of data acquisition in order to fragment the most intense precursor ions (Appendix A). However, the in-house library search may have resulted in inaccurate identification due to the presence of multiple isomers for one phenolcarboxylic or tannic compound. In addition, an accurate identification by our in-house library search also required a high resolution and a high signal-to-noise ratio of the MS and MS/MS spectra for the compounds identified.

#### 3.1.3. Identification Based on the Cleavage Law of Standard Constituents

As shown in Figure 3 and Appendix A, gallic acid (**12**), ellagic acid (**110**), and chebulic acid (**3**) yielded their characteristic ions at *m*/*z* 169.0137 [M–H]^−^, 300.9990 [M–H]^−^/ 603.0050 [2M–H]^−^, and 355.0310 [M–H]^−^/ 711.0690 [2M–H]^−^, respectively, while the known 4-*O*-galloyl-(-)-shikimic acid (**30**) and 5-*O*-galloyl-(-)-shikimic acid (**33**) both yielded their [M–H]^−^ ions at *m*/*z* 325.057 and shared exactly the same MS/MS fragmentation. Gallic acid seemed to be present in almost all phenolcarboxylic and tannic constituents of CF by the characteristic ion ([M–H]^−^) present in their MS and/or MS/MS spectra. Chebulanin (**80**), chebulagic acid (**95**), and chebulinic acid (**107**) all produced a fragment ion at *m*/*z* 337.0201 (chebuloyl fragment) by cleavage of a Che unit. This fragment would be successively dehydrated and decarboxylated in order to yield fragment ions at *m*/*z* 319.0090 and 275.0200, or decarboxylated and dehydrated to yield fragment ions at *m*/*z* 293.0303 and 275.0200. Chebulic acid also produced a fragment ion at *m*/*z* 337.0201; however, with a molecule different from the aforementioned compounds, chebuloyl could be generated by the rearrangement of chebulic acid. It should be noted that most ellagitannins generated additional ions of [M–2H]^2−^ or [2M–H]^−^, in addition to the deprotonated ion of [M–H]^−^. Their core structures could be easily identified by characteristic fragment ions with the sequential or simultaneous loss of 337 Da (chebuloyl unit). The compounds containing the HHDP moiety, such as punicalagins A (**39**) and B (**47**), corilagin (**56**), and chebulagic acid, produced a fragment ion at *m*/*z* 300.9990 in a way similar to ellagic acid [35] and compounds containing the ellagic acyl group, as exemplified by terminalin (**112**) and 4-*O*-(3″,4″-di-*O*-galloyl-*α*-l-rhamnosyl)ellagic acid (**121**). The typical [M–H]^−^ ion of the compound 1,3,6-tri-*O*-galloyl-*β*-d-glucose (**68**) could be observed at *m*/*z* 635.0879, which firstly lost a 170 Da (a gallic acyl unit) to yield *m*/*z* 465.0668, and then lost another galloyl unit and a C_2_H_2_O moiety to yield fragment ions at *m*/*z* 313.0569 and 271.0459, respectively. On this basis, a series of non-standard known and previously unknown compounds could be identified.

#### 3.1.4. Isomer and Unknown Structure Discrimination Based on Collision Cross-Sections

To date, ion mobility spectroscopy (IMS) has enabled the separation of compounds on the basis of differences in the mobility of ions through buffering gases in an electric field. IM-QTOF-MS offers respective CCS of the target analytes, which is a unique physicochemical parameter associated with the ion shape, charge, and size, as well as the compound’s structure and conformation [36]. Herein, CCS of the identified constituents were measured and predicted, as listed in Table 1, and some isomers were distinguished by comparisons of the measured CCS values with those predicted by machine-learning algorithms. For mono-galloyl glucoses, the predicted CCS values (170.416–176.058 Å^2^) showed a good match with the measured CCS values (172.066–175.413 Å^2^), but for other compounds, the CCS simulation was unsatisfactory. Yet, the galloylation (or other acylation) site on a glycose or shikimic acid, or a quinic acid could not be assigned due to the almost identical CCS values from different isomers with unidentifiable arrival times and similar structures [37]. Despite this, CCS for all the possible chemical structures deduced from the aforementioned molecular networking were predicted before the comparison with the measured results of the constituents in CF, providing a solution for refining the final identification results, to some extent.

#### 3.1.5. Global Profiling of the Phenolcarboxylic and Tannic Constituents in CF

Plant polyphenolic acids, especially hydrolyzable tannins (HTs), have been increasingly recognized as playing vital roles in long-term health due to their reduction in the risk of chronic diseases [37]. HTs possess ester and glycosidic bonds in their structures and are easily hydrolyzed into simple phenolic compounds, sugars, and polyols. Gallotannins, ellagitannins, and cheublic ellagitannins [4,38] were the main HTs of CF. The hydrolysis of gallotannins yielded gallic acid, sugar (mostly glucose), and polyols. Ellagitannins (ETs) represent a complex class of tannins formed by the esterification of polyols (i.e., glucose or rhamnose) with hexahydroxydiphenoyl (HHDP) or phenolic acid, related to its source, and have been hydrolyzed to yield ellagic acid [35]. The common phenolic acyls include valoneoyl (Val), sanguisorboyl (Sang), dehydrohexahydroxydiphenoyl (DHHDP), and chebuloyl (Che), which can be derived from HHDP by dehydrogenation, rearrangement, cycloreversion, etherification, or other reactions (Figure 2C). Due to the presence of polyhydroxy groups in sugar and polyols, ETs have an enormous structural variability due to the different linking forms between HHDP residues and glucose moiety, and, in particular, due to their tendencies to polymerize into dimeric and oligomeric derivatives or form multiple isomers after esterification with one or more functional groups [13,35]. Notably, chebublic ellagitannins appear to be the HTs with the highest content and the most complex structure in CF.

By utilizing the described integrative identification strategy, a total of 122 constituents (Table 1, Appendix A) were identified in CF. There were 106 phenolcarboxylic and tannic compounds that could be classified into four categories, including 20 phenolcarboxylic acids, 28 gallotannins, 25 ellagitannins, and 33 chebulic ellagitannins. Among them, gallic acid, ellagic acid, and chebulic acid were the most common phenolcarboxylic acids as structural units frequently used by hydrolyzable tannins of CF. Overall, CF is rich in HTs with relatively complex and diverse chemical structures, high molecular weights, and low abundance, which introduces certain difficulties into the differentiation and characterization.

As previously mentioned, MS/MS molecular networking represents a network display of MS/MS spectral data, allowing the simultaneous identification of known metabolites and their structural analogs from the crude extracts and enriched components of CMs through the clustering of similar MS/MS spectral nodes. Despite the intrinsically complex constituents, the diverse mass-spectral-fragmentation pathways, and the lack of specialized MS and MS/MS libraries dedicated to natural products, the rapid and accurate annotation of the network nodes continues to be possible and commendable in order to achieve the successful application of this method. Routinely, identification based on the cleavage law (viz. fragmentation pathways) of standard constituents has been well developed as the most reliable method for the qualitative and quantitative analysis of herbal medicines; however, its application has been limited and, at times, impractical when the standard substances for the major constituents were limited. Under these circumstances, the construction of an in-house compound library dedicated to automatic search enabled a global preliminary speculation concerning the constituents and proved to be an effective choice for the node annotation of molecular networks. In addition, CCS measurement and prediction specialized for the discrimination of isomers and unknown structures was also developed. However, due to the lack of measured CCS values of phenolcarboxylic and tannic isomers and without a well-established understanding of the definitive characteristic fragmentations from simple phenolcarboxylic molecules to those oligomerized and large polymerized tannic molecules of standard compounds, the comprehensive use of the above complementary or mutually validated methods remains imperfect and controversial for fast, accurate, and global identification of the constituents in CF.

### 3.2. UPLC-PDA Fingerprinting and Multivariate Statistical Analysis

The UPLC-PDA fingerprints of 18 batches of CF samples from different origins were generated, as shown in Figure 4A,B, and the Appendix A, and the normalized UPLC chromatograms showed satisfactory peak resolutions. Twenty-three common peaks were observed and assigned by comparing their UV spectra and retention times with those of the isolated or purchased standard compounds, namely chebulic acid (**3**), gallic acid (**12**), 1-*O*-galloyl-*β*-d-glucose (**15**), 4-*O*-galloyl-(-)-shikimic acid (**30**), 5-*O*-galloyl-(-)-shikimic acid (**33**), 3-*O*-galloyl-(-)-shikimic acid (**34**), MN-012 (**36**), punicalagin A (**39**), punicalagin B (**47**), corilagin (**56**), terflavin A (**60**), 1,3,6-tri-*O*-galloyl-*β*-d-glucose (**68**), 1,2,6-tri-*O*-galloyl-*β*-d-glucose (**72**), 3,4-di-*O*-galloylshikimic acid (**74**), chebulanin (**80**), urolithin M5 (**86**), 2,3,4,6-tetra-*O*-galloyl-*β*-d-glucose (**89**), chebulagic acid (**95**), 1,2,3,4,6-penta-*O*-galloyl-*β*-d-glucose (**97**), chebulinic acid (**107**), ellagic acid (**110**), 4-*O*-(4″-*O*-galloyl-*α*-l-rhamnosyl)ellagic acid (**118**), and 4-*O*-(3″,4″-di-*O*-galloyl-*α*-l-rhamnopyranosyl)ellagic acid (**121**) (Figure 4B). A similarity analysis was then used to evaluate the stability of the complex chemical composition and the consistency of collected samples in batches of CF. The similarities ranged from 0.637 to 0.973, and CF samples could be preliminarily divided into three groups (Appendix A): (1) Group I with the fruit samples **S1**–**S5** (0.846 < similarity < 0.904) collected from India, (2) Group II with the fruit samples **S6**–**S12** (0.924 < similarity < 0.973), and (3) Group III with the flesh of samples **S13**–**S18** (0.637 < similarity < 0.810).

Hierarchical cluster analysis (HCA) and principal component analysis (PCA) were used to distinguish CF from different sources by generating different clusters in search of the similarities between fingerprints. The results were identical with the projection suggested by the above similarity analysis. In the HCA heatmap in Figure 4C and the Appendix A, the rows represent different batches of CF while the columns represent different common constituents, the color box indicates the relative abundance of the compounds, and red represents the highest one. The CF samples fell into three main clusters related to the inherent differences in chemical composition. Cluster 1 consisted of samples **S1**–**S5,** with dominantly high contents of chebulagic acid; cluster 2 contained samples **S6**–**S12,** with favorable contents of chebulinic acid; and cluster 3 included samples **S13**–**S18,** with considerable contents of ellagic acid and gallic acid. As shown in Figure 4C, there were significant differences in the distributions of the major constituents among the CF samples that originated from different areas and/or with different medicinal parts. The PCA classification was accomplished with the first principal component (PC1) and the second principal component (PC2), which accounted for 53.7% and 31.9%, respectively, and all the samples were within the 95% confidence limit (Figure 4D). The PCA score plotting suggested the same sample classification as the fingerprint similarity analysis and HCA (Figure 4C). Notably, the degradation of the ellagitannins in the flesh of *T. chebula* during storage may have been responsible for the content increase of ellagic acid in Group III [35]. An orthogonal, partial-least-squares discriminant analysis (OPLS-DA) was used for sample discrimination and screening of the differential chemical markers for CF, with the good fit of R^2^X (0.917) and R^2^Y (0.977), and a goodness-of-prediction (Q^2^ = 0.956). The OPLS-DA score-plotting had a similar clustering result and a satisfactory separation of samples into groups II and III, which had failed to be distinguished by other multivariate statistical analyses (Figure 4E). As shown in Figure 4F, eight differential constituents, including **68** (1,3,6-tri-*O*-galloyl-*β*-d-glucose), **60** (terflavin A), **72** (1,2,6-tri-*O*-galloyl-*β*-d-glucose), **47** (punicalagin B), **107** (chebulinic acid), **95** (chebulagic acid), **97** (1,2,3,4,6-penta-*O*-galloyl-*β*-d-glucose), and **3** (chebulic acid), were discriminated based on their variable importance for the projection (VIP) values above 1.0 (Appendix A) and possessing chemotaxonomic potentials in the discrimination of these 18 batches of CF.

As depicted in Figure 4C, the CF samples were clustered into three groups with the dominant contents of chebulagic acid (**95**), chebulinic acid (**107**), and ellagic acid (**110**)/gallic acid (**12**). Chebulagic acid, chebulinic acid, and chebulic acid were screened out according to their chemotaxonomic potentials. Interestingly, seven out of the eight potential chemotaxonomic markers, as previously mentioned, belonged to the hydrolysable tannins that played crucial roles in the quality control of CF. Among the eight differential constituents, there were three gallotannins, one phenolcarboxylic acid, two ellagitannins, and two chebulic ellagitannins, most of which possessed significant antioxidant properties [4]. In addition, 1,3,6-tri-*O*-galloyl-*β*-d-glucose (**68**) showed strong binding capabilities at locations for receptor-binding domain mutants in order to inhibit the viral entry of SARS-CoV-2 into target cells [39]. A total of **95** (chebulagic acid) and **107** (chebulinic acid) showed direct and potent-dose-dependent anti-viral activities in vitro against HSV-2 and exhibited potent HCV NS3/4A inhibitory activities [8,40]. As reported, antioxidant treatment ameliorated respiratory syncytial virus-induced disease and lung inflammation, as well as potentially prevented long-term effects associated with RSV infection, such as bronchial asthma [41]. According to a recent study [9], CF showed a significant dose-dependent antioxidant effect due to its contents of hydrolyzable tannins. Herein, we found that the contents of hydrolyzable tannins in CF samples from group III were apparently lower than the samples from the other two groups.

### 3.3. Antioxidant Evaluation and Screening of the Antioxidant Constituents

DPPH^•^-radical-scavenging assay is one of the most common methods used for antioxidant evaluation in vitro, while ABTS^+^ and FRAP assays are combined regularly to investigate antioxidant potentials. In our study, as shown in Appendix A, CF sample **S5** exhibited the most potent antioxidant activities with respective values of 2.993 μg/mL (IC_50_ of DPPH^•^ radical scavenging), 14.47 ± 1.106 mmol/g (ABTS+ radical scavenging capacity), and 1.566 ± 0.182 mmol/g (ferric reducing antioxidant power), in the three assays. The DPPH^•^ radical-scavenging IC_50_ values of samples **S1**–**S12** (2.993 ≤ IC_50_ ≤ 5.303 μg/mL) were significantly lower than the samples **S13**–**S18** (5.761 ≤ IC_50_ ≤ 10.110 μg/mL), and we noticed that samples **S1**–**S12** were all from the whole fruits of CF, while samples **S13**–**S18** were all from the flesh of CF. This phenomenon indicated that whole-fruit preservation before processing and extraction was beneficial for the conservation of the active antioxidant ingredients in CF, such as chebulagic acid and chebulinic acid.

The spectrum-effect correlation is a reliable method to understanding the relationship between the efficacy and the constituents using chemometric methods. Herein, Pearson’s correlation analysis and the grey relational analysis were performed for the spectrum–antioxidant effect correlation between the UPLC fingerprints and the antioxidant effects.

A heat-map of Pearson’s correlation coefficients was plotted of the normalized 23 common peak areas (Figure 5A), the IC_50_ values of the DPPH^•^ assay, and the total antioxidation capabilities (T-AOC) of the ABTS^+^ and FRAP assays for the 18 batches of CFME. The dark red and blue points depict the positive and negative correlations, respectively. Generally, the DPPH^•^ radical-scavenging IC_50_ values were negatively correlated to the antioxidant activity while the T-AOC values in the ABTS^+^ and FRAP assays were positively correlated to the antioxidant activity. The correlation coefficients of the ABTS^+^ and FRAP assays were exactly opposite of those of the DPPH^•^ assay. As a result, seven peaks (**60**, **72**, **86**, **89**, **95**, **97**, and **107**) with r < –0.5 showed obviously negative correlations to the DPPH^•^ radical-scavenging activity, while six peaks (**60**, **72**, **89**, **95**, **97**, and **107**) with r > 0.5 exhibited significantly positive correlations to the ABTS^+^ radical-scavenging and FRAP activities. Six antioxidant-contributing constituents, represented by peaks **60** (terflavin A), **72** (1,2,6-tri-*O*-galloyl-*β*-d-glucose), **89** (2,3,4,6-tetra-*O*-galloyl-*β*-d-glucose), **95** (chebulagic acid), **97** (1,2,3,4,6-penta-*O*-galloyl-*β*-d-glucose), and **107** (chebulinic acid), stood out due to having significantly negative correlations with the DPPH^•^ radical-scavenging activity and significantly positive correlations with the ABTS^+^ radical-scavenging and FRAP activities. Notably, the relative contents of the large molecular tannins were negatively correlated to those of the small molecular phenolcarboxylic units, further indicating the existence of intrinsic degradation among the large molecular tannins with potent antioxidant abilities in the CF samples.

The grey relational analysis (GRA) was performed on the relative contents of the UPLC chromatographic peaks and the antioxidant capabilities, and the distances between the variables were calculated using the mean values to obtain the correlations between the contents (%) of the constituents and the antioxidant activities of the CF methanol extracts (Figure 5B). As a result, the relational degrees (r) between the identified constituents and the antioxidant efficacies ranged from 0.5289 to 0.8845. Specifically, there were 8 constituents (**15**, **39**, **56**, **74**, **80**, **89**, **95**, and **97**) with grey relational degrees above 0.7, indicating that the antioxidant capacity of CF may have been the overall effects of those dedicated constituents with high relevance.

Finally, a comparative UPLC analysis, before and after treatment with DPPH^•^ [30], was carried out to verify the main antioxidant metabolites in CF. As shown in Figure 5C, almost all the main constituents, except for the triterpenoid constituents, exhibited varying degrees of antioxidant activities indicated by the significant decrease in the relative contents, represented by the decrease (%) in their UPLC peak areas after treatment with DPPH^•^ (Appendix A). The results showed that most of the active constituents were polymeric phenolcarboxylic or tannic compounds, and **47** (punicalagin B), **72** (1,2,6-tri-*O*-galloyl-*β*-d-glucose), **56** (corilagin), **95** (chebulagic acid), and **107** (chebulinic acid) possessed the most potent DPPH radical-scavenging activities, with decreases of 43.16%, 31.48%, 31.06%, 27.18%, and 22.01% in their peak areas, respectively. Interestingly, the monomeric phenolcarboxylic or tannic compounds, the so-called structural units of tannins, such as **110** (ellagic acid), **12** (gallic acid), and **3** (chebulic acid), showed much weaker DPPH^•^ scavenging effects, with decreases of 8.69%, 8.31%, and 4.19% in their peak areas, respectively, even though they had been previously thought to show excellent antioxidant activities [42].

As reported, the 13 major constituents demonstrated potent antioxidant activities, and the CF extracts showed stronger antioxidant activities than the individual chemical constituents [4]. However, our spectrum-effect correlation and the UPLC-DPPH^•^ analysis revealed that the main phenolcarboxylic and tannic compounds, especially the so-called structural units of tannins, including chebulic acid and gallic acid, did not exert a favorable antioxidant contribution to the overall antioxidant activity of CFME. This phenomenon indicated that there may exist synergistic or antagonistic effects on the antioxidant activities of the main constituents in the crude extract of CF. Finally, our study resulted in the discoveries of compounds **15** (1-*O*-galloyl-*β*-d-glucose), **39** (punicalagin A), **56** (corilagin), **60** (terflavin A), **72** (1,2,6-tri-*O*-galloyl-*β*-d-glucose), **74** (3,4-di-*O*-galloylshikimic acid), **80** (chebulanin), **89** (2,3,4,6-tetra-*O*-galloyl-*β*-d-glucose), **95** (chebulagic acid), **97** (1,2,3,4,6-penta-*O*-galloyl-*β*-d-glucose), and **107** (chebulinic acid) significantly indicated the antioxidant potential of CF extracts by both Pearson’s correlation analysis and a grey relational analysis. In addition, compounds **47** (punicalagin B), **56** (corilagin), **72** (1,2,6-tri-*O*-galloyl-*β*-d-glucose), **95** (chebulagic acid), and **107** (chebulinic acid) exhibited potent antioxidant activities as verified by UPLC-DPPH^•^ analysis.

Overall, the characterization and the fingerprinting of the phenolcarboxylic and tannic constituents, the antioxidant evaluation, and the spectrum-effect correlation were accomplished successively in our study, enabling the development of antioxidant constituent markers for chemotaxonomic significance when distinguishing CF samples. At the same time, we speculated that the common phenolcarboxylic and tannic constituents with high structural similarities possibly undergo mutual transformation and/or degradation during collection, preservation, and processing prior to being prescribed in a clinical setting, which will benefit from our ongoing in-depth studies and research.

## 4. Conclusions

In summary, with the specialized integrative identification strategy, a total of 122 constituents of CF were identified, including 20 phenolcarboxylic acids, 28 gallotannins, 25 ellagitannins, and 33 chebulic ellagitannins, and among them 38 were identified for the first time as new compounds of CF. Furthermore, 1,3,6-tri-*O*-galloyl-*β*-d-glucose, terflavin A, 1,2,6-tri-*O*-galloyl-*β*-d-glucose, punicalagin B, chebulinic acid, chebulagic acid, 1,2,3,4,6-penta-*O*-galloyl-*β*-d-glucose, and chebulic acid, among the 23 common characteristic constituents, of the CF samples were screened out after multivariate statistical analyses. Finally, the antioxidant capacity of CF was evaluated, and the spectrum–antioxidant correlation was determined by Pearson’s correlation analysis and a grey relational analysis before a comparative DPPH^•^-pre-treated UPLC analysis was applied to verify the significant antioxidant metabolites in CF. Our study not only provided a promising strategy for the global characterization and identification of the antioxidant constituents of CF, but it also laid a good technical and methodological foundation for future quality control and applications of CF. However, future studies are still needed to uncover the composition changes during collection, preservation, and processing; the distribution of phenolcarboxylic and tannic constituents in CF; the easily confused CMs; the different medicinal parts of *T. chebula*; and the development of medicinal and industrial products. In addition, in vivo and clinical applications of CF antioxidants are still needed before the development and application of CF and its main metabolites can be fully realized.

## Figures and Tables

**Figure 1 antioxidants-12-02093-f001:**
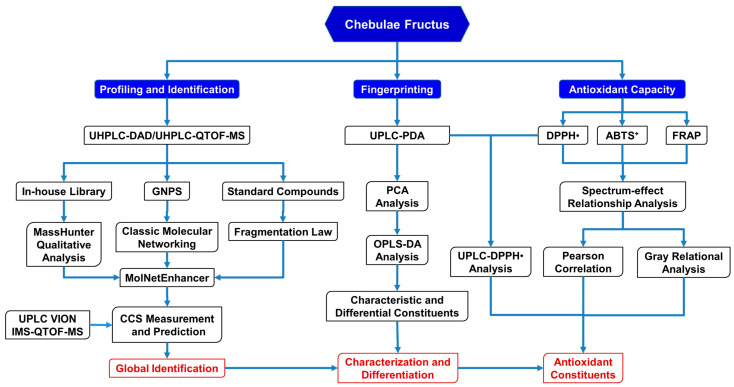
The new strategy for identification and characterization of the components in CF.

**Figure 2 antioxidants-12-02093-f002:**
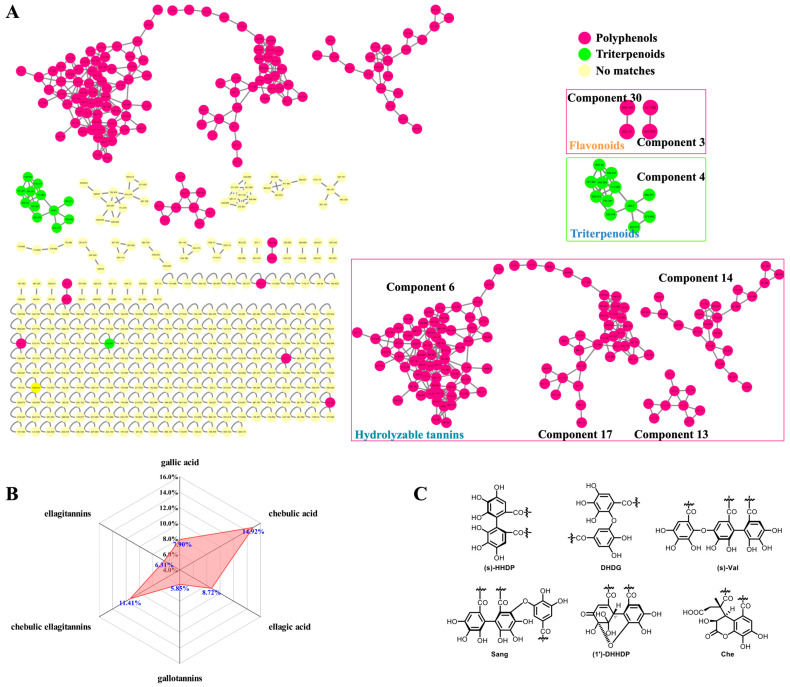
Classic molecular networking re-analyzed by MolNetEnhancer. (**A**) GNPS-based molecular networking of the UHPLC-QTOF-MS data; (**B**) the display of the ion abundance ratios of three types of hydrolyzable tannins and their core phenolcarboxylic acyls in mass spectrometry by spider-web mode; (**C**) the common phenolcarboxylic acyls for HHDP.

**Figure 3 antioxidants-12-02093-f003:**
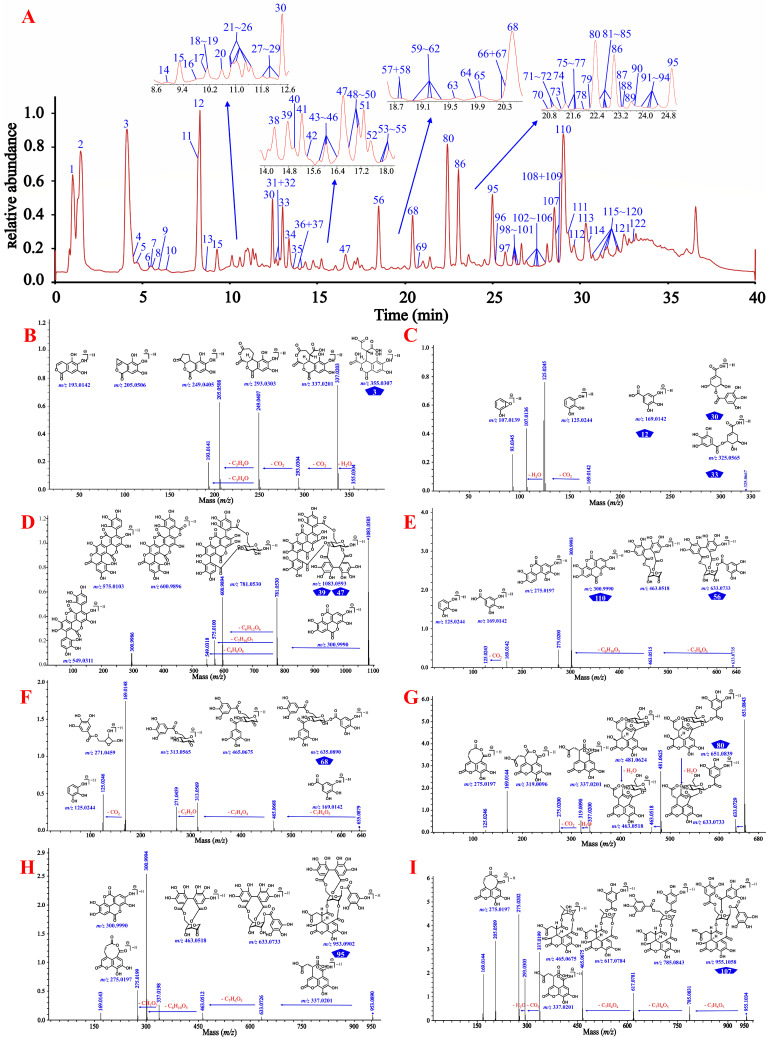
The total ion current chromatogram and the cleavage law of standard compounds in mass spectrometry. (**A**) 122 compounds detected in the extract of CF by UHPLC-QTOF-MS analysis. (**B**) the cleavage law of chebulic acid (**3**); (**C**) the cleavage law of gallic acid (**12**), 4-*O*-galloyl-(-)-shikimic acid (**33**), and 5-*O*-galloyl-(-)-shikimic acid (**34**); (**D**) the cleavage law of punicalagins A (**39**) and B (**47**); (**E**) the cleavage law of corilagin (**56**) and ellagic acid (**110**); (**F**) the cleavage law of 1,3,6-tri-*z*-galloyl-*β*-d-glucose (**68**); (**G**) the cleavage law of chebulanin (**80**); (**H**) the cleavage law of chebulagic acid (**95**); (**I**) the cleavage law of chebulinic acid (**107**); (**J**) the cleavage law of terminalin (**112**); (**K**) the cleavage law of 4-*O*-(3″,4″-di-*O*-galloyl-*α*-l-rhamnosyl)ellagic acid (**121**).

**Figure 4 antioxidants-12-02093-f004:**
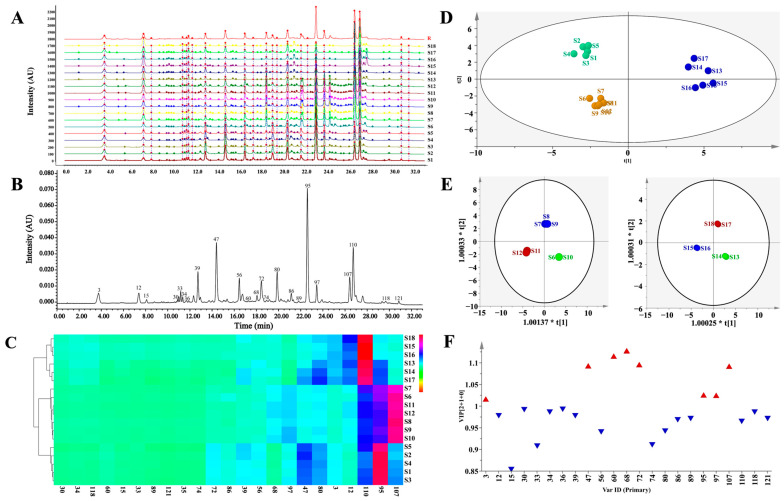
Chromatographic fingerprinting and multi-variate statistical analysis of CF samples. (**A**) UPLC fingerprints of 18 batches of CF; (**B**) the representative UPLC fingerprints with 23 common peaks marked; (**C**) the heat-map of 18 batches of CF and the 23 constituents; (**D**) score-plotting of the principal component analysis results for UPLC-DAD data obtained for CF; (**E**) score-plotting of the orthogonal partial least square discriminant analysis for the Group II and III; (**F**) VIP histogram of the main constituents (red: VIP > 1; blue: VIP < 1).

**Figure 5 antioxidants-12-02093-f005:**
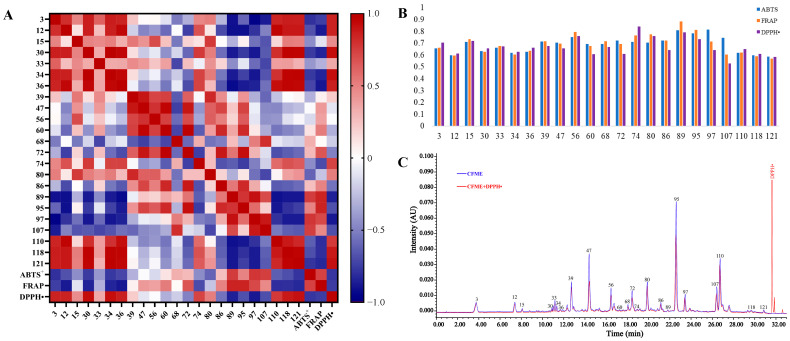
Modeling of the spectrum–antioxidant effect relationship and UPLC-DPPH^•^ analysis for CF. (**A**), Pearson’s correlation heatmap; (**B**), GRA correlation graphs; (**C**), representative chromatogram of the UPLC-DPPH^•^ analysis.

**Table 1 antioxidants-12-02093-t001:** The compounds identified by UPLC/IM-QTOF-MS and MS/MS molecular networking.

No.	Observed *t*_R_ (min)	Formula	Identification	Measured CCS (Å²)	Predicted CCS (Å²)	Observed Mass (Da)	Mass Error (ppm)	Quasi-Molecular /Adducts Ion	MS/MS Fragmentation
**1**	1.04	C_7_H_11_O_6_	quinic acid	–	142.889	191.0564	4.2	[M–H]^−^	85.0297, 93.0343, 59.0141, 55.0193, 67.0191, 108.0214
**2**	1.52	C_7_H_9_O_5_	shikimic acid	–	135.297	173.0454	2.3	[M–H]^−^	93.0344, 65.0399, 55.0194
**17**	10.11	C_11_H_11_O_8_	MN-01	–	155.806	271.0454	0.0	[M–H]^−^	124.0167, 78.0113
phenolcarboxylic acids									
**3** *	4.15	C_14_H_11_O_11_	chebulic acid	–	174.904	355.0329	7.9	[M–H]^−^	149.0244, 193.0141, 164.0475, 135.0452, 164.0475, 179.0707, 107.0497
**10**	6.32	C_17_H_17_O_13_	MN-02	189.370	197.766	429.0678	2.1	[M–H]^−^	193.0142, 163.0395, 205.0503, 133.0657, 151.0398, 59.0140, 249.0405
**12** *	8.24	C_7_H_5_O_5_	gallic acid	124.174	126.339	169.0145	4.7	[M–H]^−^	125.0244, 169.0143, 79.0184
**27**	11.85	C_16_H_15_O_11_	MN-03	179.820	185.931	383.0617	0.8	[M–H]^−^	193.0138, 205.0508, 163.0399, 135.0450, 233.0062
**30** *	12.44	C_14_H_13_O_9_	4-*O*-galloyl-(-)-shikimic acid	173.481	173.304	325.0575	4.6	[M–H]^−^	125.0168, 93.0344, 169.0144
**33** *	13.01	C_14_H_13_O_9_	5-*O*-galloyl-(-)-shikimic acid	175.504	174.817	325.0571	3.4	[M–H]^−^	125.0244, 107.0137, 93.0344, 169.0138
**34**	13.38	C_14_H_13_O_9_	3-*O*-galloyl-(-)-shikimic acid	174.327	175.075	325.0572	3.7	[M–H]^−^	125.0246, 107.0136, 93.0343, 169.0142
**42**	15.38	C_12_H_11_O_7_	MN-04	158.510	159.938	267.0512	2.6	[M–H]^−^	247.9025, 166.8641, 124.0166, 107.0133, 78.0114
**45**	16.08	C_9_H_9_O_5_	ethyl gallate	–	141.145	197.0461	5.6	[M–H]^−^	124.0166, 78.0155, 51.0244
**50**	17.14	C_14_H_9_O_9_	digallic acid	–	166.869	321.0254	2.2	[M–H]^−^	125.0244, 169.0146, 79.0188
**52**	17.52	C_21_H_9_O_13_	flavogallonic acid	–	199.719	469.0055	2.6	[M–H]^−^	299.9913, 379.0090, 407.0039, 300.9979, 351.0145, 335.0195, 425.0136
**58**	18.78	C_21_H_17_O_13_	3,5-di-*O*-galloylshikimic acid	203.970	214.557	477.0675	1.3	[M–H]^−^	125.0244, 93.0343, 123.0083, 263.0552, 201.0547, 169.0143
**64**	19.95	C_13_H_7_O_8_	brevifolin carboxylic acid	152.493	160.170	291.0147	2.1	[M–H]^−^	145.0293, 190.0268, 247.0238
**67**	20.40	C_21_H_17_O_13_	4,5-di-*O*-galloylshikimic acid	198.260	210.711	477.0674	1.0	[M–H]^−^	169.0143, 125.0244, 201.0547, 93.0343, 51.0245
**74**	21.41	C_21_H_17_O_13_	3,4-di-*O*-galloylshikimic acid	198.420	212.094	477.0681	2.5	[M–H]^−^	169.0143, 125.0244, 263.0552, 201.0547, 93.0343, 51.0245
**82**	22.70	C_11_H_13_O_5_	MN-05	153.340	151.904	225.0762	–0.4	[M–H]^−^	153.7419, 124.0167, 78.0113
**86** *	23.03	C_13_H_7_O_7_	urolithin M5	149.257	153.132	275.0199	2.5	[M–H]^−^	145.0296, 173.0243, 117.0342, 229.0139
**94**	24.86	C_28_H_21_O_17_	3,4,5-tri-*O*-galloylshikimic acid	230.001	241.753	629.0790	1.7	[M–H]^−^	289.0339, 245.0439, 201.0551, 169.0143, 125.0243, 93.0344
**110** *	29.01	C_14_H_5_O_8_	ellagic acid	153.266	160.297	300.9995	3.7	[M–H]^−^	145.0294, 117.0344, 200.0117, 172.0169, 283.9964, 173.0247, 129.0348, 201.018, 299.9915, 300.9988
**112** *	29.64	C_28_H_9_O_16_	terminalin	–	222.392	600.9900	1.5	[M–H]^−^	298.9832, 270.9880, 300.9989, 299.9897, 485.0143, 582.9772
gallotannins									
**4**	4.49	C_13_H_15_O_10_	6-galloyl-*β*-d-glucose	172.592	171.292	331.0667	0.6	[M–H]^−^	169.0144, 125.0243, 107.0134, 59.0143, 51.0229
**7**	5.47	C_13_H_15_O_10_	4-galloyl-*β*-d-glucose	173.955	174.532	331.0669	1.2	[M–H]^−^	125.0242, 169.0141, 107.0134, 59.0140
**15** *	9.27	C_13_H_15_O_10_	1-*O*-galloyl-*β*-d-glucose	172.066	170.416	331.0669	1.2	[M–H]^−^	59.0142, 169.0139, 125.0243, 107.0136, 51.0245
**18**	10.12	C_13_H_15_O_10_	3-galloyl-*β*-d-glucose	175.413	175.534	331.0669	1.2	[M–H]^−^	125.0243, 107.0134, 169.0140, 59.0143
**19**	10.12	C_13_H_17_O_10_	MN-06	175.750	174.430	333.0830	2.4	[M–H]^−^	125.0244, 107.0136, 169.0141, 59.0141
**22**	10.85	C_13_H_15_O_10_	2-galloyl-*β*-d-glucose	–	–	331.0673	2.4	[M–H]^−^	169.0137, 125.0243, 107.0134, 59.0143, 211.0248
**28**	12.05	C_20_H_19_O_14_	2,4-di-*O*-galloyl-d-glucose	198.512	209.402	483.0780	1.0	[M–H]^−^	169.0143, 125.0245, 313.0573, 271.0453, 211.0240
**35**	13.47	C_20_H_19_O_14_	2,3-di-*O*-galloyl-d-glucose	197.150	206.773	483.0780	1.0	[M–H]^−^	169.0140, 125.0243, 313.0564, 271.0453, 211.0248
**38**	14.32	C_20_H_19_O_14_	1,2-di-*O*-galloyl-d-glucose	195.054	202.905	483.0780	1.0	[M–H]^−^	169.0144, 125.0245, 271.0452, 211.0245, 59.0142
**41**	15.23	C_20_H_19_O_14_	1,3-di-*O*-galloyl-d-glucose	195.827	204.454	483.0780	1.0	[M–H]^−^	169.0145, 125.0246, 271.0454, 211.0246
**46**	16.37	C_20_H_19_O_14_	1,6-di-*O*-galloyl-d-glucose	194.986	200.090	483.0780	1.0	[M–H]^−^	169.0142, 125.0243, 313.0562, 271.0449, 211.0246
**48**	16.74	C_20_H_19_O_14_	4,6-di-*O*-galloyl-d-glucose	196.386	205.002	483.0786	2.3	[M–H]^−^	169.0142, 125.0241, 313.0544, 271.0455, 211.0240
**49**	17.10	C_20_H_19_O_14_	3,4-di-*O*-galloyl-d-glucose	195.162	203.345	483.0790	3.1	[M–H]^−^	169.0143, 125.0244, 211.0244, 271.0453, 313.0573
**53**	17.93	C_20_H_19_O_14_	3,6-di-*O*-galloyl-d-glucose	196.961	205.228	483.0779	0.8	[M–H]^−^	169.0143, 125.0243, 313.0573, 271.0453, 211.0240, 423.0000, 151.0032
**55**	18.13	C_27_H_23_O_18_	tri-*O*-galloyl-*β*-d-glucose	226.655	–	635.0898	2.2	[M–H]^−^	169.0144, 465.0668, 423.0566, 313.0567, 211.0243, 271.0453
**63**	19.58	C_27_H_23_O_18_	tri-*O*-galloyl-*β*-d-glucose	225.329	–	635.0901	2.7	[M–H]^−^	169.0144, 125.0244, 635.0894, 483.0763, 423.0566, 313.0567, 211.0243
**66**	20.26	C_19_H_19_O_11_	MN-07	201.510	196.492	423.0928	0.2	[M–H]^−^	169.0136, 109.0289, 125.0244, 203.0373, 299.0122
**68** *	20.44	C_27_H_23_O_18_	1,3,6-tri-*O*-galloyl-*β*-d-glucose	229.419	229.367	635.0888	0.6	[M–H]^−^	169.0144, 313.0567, 211.0243, 635.0894, 465.0668, 271.0461
**69**	20.69	C_27_H_23_O_18_	tri-*O*-galloyl-*β*-d-glucose	229.174	–	635.0889	0.8	[M–H]^−^	169.0141, 125.0243, 313.0559, 211.0243, 635.0894, 465.0654, 271.0461, 423.0572
**70**	20.83	C_27_H_23_O_18_	1,4,6-tri-*O*-galloyl-*β*-d-glucose	228.727	228.651	635.0895	1.7	[M–H]^−^	169.0142, 125.0242, 313.0562, 211.0239, 465.0660, 271.0448, 423.0561
**72**	21.05	C_27_H_23_O_18_	1,2,6-tri-*O*-galloyl-*β*-d-glucose	227.836	227.577	635.0894	1.6	[M–H]^−^	169.0142, 125.0242, 313.0564, 211.0235, 465.0660
**75**	21.51	C_34_H_27_O_22_	1,2,4,6-tetra-*O*-galloyl-*β*-d-glucose	252.633	255.173	787.1000	0.8	[M–H]^−^	169.0141, 465.0659, 295.0447, 635.0876, 423.0552, 211.0246, 125.0242
**89**	23.63	C_34_H_27_O_22_	2,3,4,6-tetra-*O*-galloyl-*β*-d-glucose	266.444	259.406	787.1004	1.3	[M–H]^−^	169.0146, 465.0675, 295.0455, 125.024, 617.0776
**92**	24.17	C_34_H_27_O_22_	1,2,3,6-tetra-*O*-galloyl-*β*-d-glucose	253.240	253.666	787.0995	0.1	[M–H]^−^	169.0141, 465.0662, 617.0768, 295.0446
**97**	25.70	C_41_H_31_O_26_	1,2,3,4,6-penta-*O*-galloyl-*β*-d-glucose	296.630	282.841	939.1114	1.1	[M–H]^−^	169.0143, 617.0779, 769.0885, 447.0563, 725.0979, 295.0451
**100**	26.31	C_34_H_27_O_23_	MN-08	259.230	258.547	803.0947	0.5	[M–H]^−^	275.0196, 205.0503, 169.0144, 337.0198, 651.0861
**105**	27.70	C_29_H_25_O_15_	di-*O*-galloyl-2-*O*-cinnamoyl-*β*-d-glucose	–	239.887	613.1182	–1.8	[M–H]^−^	169.0144,125.0243, 71.0140, 21.0239, 401.0860, 443.0983
**106**	28.04	C_29_H_25_O_15_	di-*O*-galloyl-3-*O*-cinnamoyl-*β*-d-glucose	–	238.022	613.1186	–1.1	[M–H]^−^	169.0140, 125.0240, 451.0789, 211.0221, 71.0136
ellagitannins									
**6**	5.43	C_20_H_17_O_14_	MN-09	202.370	200.811	481.0620	0.4	[M–H]^−^	300.9985, 275.0197, 229.0142
**11**	8.16	C_20_H_17_O_14_	MN-10	201.750	200.377	481.0620	0.4	[M–H]^−^	300.9984, 275.0197, 229.0147, 185.0244, 125.0239
**29**	12.32	C_20_H_15_O_13_	MN-11	196.640	200.247	463.0516	0.6	[M–H]^−^	300.9990, 275.0204
**31**	12.65	C_34_H_21_O_22_	punicalin A	247.216	240.735	781.0539	1.9	[M–H]^−^	600.9889, 298.9831, 575.0083, 300.9989, 721.0299, 275.02, 448.9785, 781.0520
**32**	12.84	C_34_H_21_O_22_	punicalin B	248.928	240.735	781.0535	1.4	[M–H]^−^	600.9896, 298.9831, 575.0039, 721.0300, 781.0514
**36**	13.86	C_34_H_23_O_23_	MN-12	242.080	246.089	783.0672	–1.1	[M–H]^−^	300.9979, 275.0199, 783.0642, 481.0663
**37**	14.11	C_27_H_21_O_18_	corilagin isomer	–	223.714	633.0738	1.6	[M–H]^−^	300.9987, 275.0202, 169.0144, 463.0517
**39** *	14.77	C_48_H_27_O_30_	punicalagin A	305.520	295.162	1083.0608	1.9	[M–H]^−^	1083.0599, 600.9898, 781.0530, 575.0098, 721.0330, 300.9986
**43**	15.80	C_34_H_23_O_23_	terflavin B	241.867	249.874	783.0679	–0.3	[M–H]^−^	450.9940, 631.0549, 425.0126, 300.997, 783.0695, 169.0143
**44**	16.03	C_34_H_25_O_22_	tercatain or its isomer	267.387	247.710	785.0847	1.3	[M–H]^−^	300.9987, 275.0198, 249.0406, 633.0736, 169.0143, 785.0817
**47** *	16.59	C_48_H_27_O_30_	punicalagin B	303.497	295.162	1083.0608	1.9	[M–H]^−^	1083.0599, 600.9898, 781.0523, 721.0313, 300.9987, 549.0295, 448.9775, 249.0400
**56** *	18.49	C_27_H_21_O_18_	corilagin	224.508	224.752	633.0739	1.7	[M–H]^−^	300.9996, 275.0202, 463.0517, 169.0144
**57**	18.66	C_34_H_25_O_22_	tercatain or its isomer	268.495	249.881	785.0837	0.0	[M–H]^−^	300.9984, 275.0192, 633.0736, 615.0593, 249.0406, 169.0137
**60**	19.22	C_48_H_29_O_30_	terflavin A or its isomer	312.025	304.291	1085.0742	–0.2	[M–H]^−^	450.9933, 783.0668, 1085.0719, 933.0632, 631.0559, 425.0149, 300.9976
**61**	19.27	C_34_H_25_O_22_	tercatain or its isomer	268.495	249.063	785.0851	1.8	[M–H]^−^	300.9987, 275.0198, 633.0736, 249.0406, 169.0141
**65**	19.96	C_34_H_25_O_22_	tellimagradin I	264.090	255.565	785.0831	–0.8	[M–H]^−^	300.9987, 275.0198, 633.0723, 169.0141
**71**	21.02	C_48_H_29_O_30_	terflavin A or its isomer	311.474	305.606	1085.0747	0.3	[M–H]^−^	450.9933, 783.0668, 1085.0719, 933.0632, 631.0559, 425.0149, 300.9976
**83**	22.73	C_41_H_29_O_26_	tellimagrandin II	295.841	277.927	937.0940	–0.7	[M–H]^−^	300.9985, 275.0195, 937.0922, 767.0694, 599.0685, 465.0654, 419.0603, 169.0138
**101**	26.53	C_20_H_13_O_14_	MN-13	198.090	201.575	477.0348	9.0	[M–H]^−^	300.9988, 125.024, 302.002, 169.014
**108**	28.57	C_21_H_15_O_14_	MN-14	208.100	210.883	491.0460	–0.4	[M–H]^−^	299.9907, 270.9857, 169.0138, 125.0242
**111** *	29.32	C_20_H_15_O_12_	eschweilenol C	200.297	197.035	447.0571	1.6	[M–H]^−^	299.9941, 300.9978, 271.9950
**116**	31.12	C_21_H_17_O_12_	MN-15	205.440	202.894	461.0736	3.5	[M–H]^−^	299.9917, 315.0143, 270.9882
**118**	31.54	C_27_H_19_O_16_	4-*O*-(4″-*O*-galloyl-*α*-l-rhamnosyl)ellagic acid	213.728	228.827	599.0686	2.2	[M–H]^−^	300.9989, 169.0141, 125.0242
**121** *	32.49	C_34_H_23_O_20_	4-*O*-(3″,4″-di-*O*-galloyl-*α*-l-rhamnosyl)ellagic acid	238.780	253.167	751.0789	0.8	[M–H]^−^	300.9986, 169.0138, 449.0716
**122**	32.99	C_34_H_23_O_20_	4-*O*-(2″,4″-di-*O*-galloyl-*α*-l-rhamnosyl)ellagic acid	244.308	253.300	751.0783	0.0	[M–H]^−^	300.9986, 169.0138, 449.0716, 599.0658
chebulic ellagitannins									
**5**	4.71	C_20_H_23_O_16_	MN-16	205.480	206.812	519.0996	1.9	[M–H]^−^	205.0508, 163.0398, 193.0141, 177.0551, 133.0657, 187.0397, 249.0400, 293.0317, 337.0199
**8**	5.47	C_20_H_23_O_16_	MN-17	205.220	205.690	519.1001	1.9	[M–H]^−^	205.0510, 163.0399, 193.0141, 133.0657, 187.0397, 249.0403, 293.0303, 337.0195
**9**	5.96	C_20_H_21_O_16_	MN-18	–	207.783	517.0844	2.7	[M–H]^−^	205.0507, 193.0139, 163.0400, 133.0656, 177.0552, 249.0405, 337.0188, 293.0305, 275.0186
**13**	8.58	C_20_H_23_O_16_	MN-19	203.200	204.232	519.0997	2.1	[M–H]^−^	205.0506, 193.0140, 161.0605, 231.0290, 237.0030, 249.04, 59.0144, 293.0301, 337.0190, 401.0722, 275.0188
**14**	8.88	C_20_H_21_O_16_	MN-20	–	209.049	517.0833	0.6	[M–H]^−^	237.0292, 205.0506, 249.0401, 279.0143, 309.0259, 339.0357, 161.0607, 133.0655
**16**	9.81	C_20_H_19_O_15_	MN-21	203.370	205.632	499.0726	0.4	[M–H]^−^	163.0401, 193.0131, 151.0382, 231.0302, 205.0506, 177.0544
**20**	10.59	C_21_H_19_O_15_	MN-22	207.020	211.191	511.0739	2.9	[M–H]^−^	193.0141, 205.0506, 163.0398, 161.0608, 133.0656, 187.0398, 249.0401, 337.0206, 275.0192
**21**	10.83	C_20_H_19_O_15_	MN-23	200.635	206.525	499.0727	0.6	[M–H]^−^	193.0141, 205.0510, 231.0277, 161.0607, 187.0388
**23**	10.90	C_21_H_19_O_15_	MN-24	–	212.003	511.0736	2.3	[M–H]^−^	205.0505, 193.0141, 163.0395, 133.0652, 187.0395, 249.0402, 337.0183, 59.0142, 293.0318, 177.0546
**24**	11.02	C_21_H_19_O_15_	MN-25	–	212.647	511.0738	2.7	[M–H]^−^	205.0505, 193.0140, 163.0398, 133.0655, 187.0393, 249.0398, 337.0192, 393.0455
**25**	11.32	C_21_H_19_O_15_	MN-26	–	215.118	511.0739	2.9	[M–H]^−^	193.0141, 205.0506, 163.0399, 187.0400, 337.0190, 151.0399, 133.0658
**26**	11.49	C_21_H_19_O_15_	MN-27	–	216.331	511.0739	2.9	[M–H]^−^	193.0139, 205.0507, 163.0399, 177.0550, 187.0396, 337.0191, 151.0397, 133.0655
**40**	14.99	C_27_H_25_O_20_	phyllanemblinin E	233.848	233.045	669.0955	2.4	[M–H]^−^	205.0510, 249.0407, 193.0142, 161.0609, 337.0197, 293.0301, 133.0656
**51**	17.30	C_28_H_27_O_20_	1′-*O*-methyl neochebulanin	–	239.776	683.1136	5.9	[M–H]^−^	169.0148, 125.0248, 231.0304, 351.0357, 275.0206, 409.0773
**54**	17.95	C_34_H_29_O_24_	MN-28	262.480	258.372	821.1049	0.0	[M–H]^−^	337.0191, 483.0759, 293.0314, 249.0394, 205.0511
**59**	19.04	C_41_H_31_O_28_	neochebulagic acid	284.153	281.548	971.1003	0.1	[M–H]^−^	633.0728, 337.0198, 300.9988, 463.0514, 419.0610, 249.0404, 205.0511
**62**	19.41	C_28_H_27_O_20_	1′-*O*-methyl neochebulanin isomer	–	240.401	683.1100	0.6	[M–H]^−^	169.0142, 125.0243, 203.0341, 437.0755, 381.0462, 337.0555
**73**	21.26	C_42_H_33_O_28_	methyl neochebulagate	306.796	288.119	985.1148	–1.0	[M–H]^−^	633.0719, 783.0664, 463.0505, 351.0345, 300.9983, 231.0296, 169.0143
**76**	21.74	C_41_H_33_O_28_	neochebulinic acid	285.501	286.651	973.1167	0.9	[M–H]^−^	635.0887, 337.0194, 465.0652, 249.0396, 293.0292, 205.0495
**77**	21.75	C_33_H_35_O_24_	MN-29	275.960	258.659	815.1520	0.2	[M–H]^−^	169.0141, 435.0562, 645.1298, 205.0505, 125.0242, 381.0453, 463.0505
**78**	22.04	C_33_H_33_O_24_	MN-30	257.675	257.674	813.1367	0.6	[M–H]^−^	169.0140, 125.0244, 435.05561, 643.1137, 583.0917, 523.0714, 481.0608, 381.0453, 331.0661, 231.0296,
**79**	22.27	C_42_H_33_O_28_	6′-*O*-methyl neochebulagate	307.098	288.320	985.1152	–0.6	[M–H]^−^	300.9983, 783.0664, 633.0719, 463.0505, 351.0345,231.0296, 169.0143, 953.0876
**80** *	22.41	C_27_H_23_O_19_	chebulanin	230.233	229.479	651.0855	3.2	[M–H]^−^	169.0144, 125.0245, 275.0196, 203.0344, 337.0560, 409.0770
**81**	22.58	C_42_H_35_O_28_	1′-methyl neochebulinate	–	292.165	987.1302	–1.3	[M–H]^−^	635.0885, 465.0654, 351.0345, 169.0142
**84**	22.75	C_41_H_33_O_29_	MN-18	299.280	288.344	989.1107	–0.1	[M–H]^−^	337.02, 249.04, 205.05, 338.023, 293.03, 193.013, 250.043, 163.039, 161.061, 339.025, 275.018, 481.06, 177.055, 133.065, 319.008, 381.045, 437.071, 483.063, 499.071
**85**	22.93	C_43_H_37_O_28_	dimethyl neochebulinate	290.933	298.271	1001.1469	–0.2	[M–H]^−^	635.0885, 465.0654, 365.0502, 169.0140
**88**	23.34	C_34_H_31_O_23_	MN-31	256.171	258.627	807.1259	0.4	[M–H]^−^	169.0144, 275.0193, 481.0611, 435.0552, 231.0290, 637.1017, 125.02433
**90**	23.75	C_43_H_35_O_28_	dimethyl neochebulagate or dimethyl 4’-*epi*-neochebulagate	290.933	293.830/293.873	999.1318	0.3	[M–H]^−^	300.9985, 275.0190, 205.0505, 829.1083, 633.0756, 527.1011, 463.0513, 365.0506
**91**	23.92	C_41_H_33_O_29_	MN-32	297.870	286.938	989.1112	0.4	[M–H]^−^	337.0197, 249.0401, 205.0507, 293.0312
**95** *	24.97	C_41_H_29_O_27_	chebulagic acid	301.139	274.834	953.0911	1.6	[M–H]^−^	300.9990, 275.0198, 205.0504, 337.0199, 169.0144, 463.0513, 633.0724
**96**	25.12	C_34_H_27_O_23_	MN-33	259.196	253.905	803.0949	–2.1	[M–H]^−^	169.0143, 275.0201, 481.0621, 633.0727, 293.0283, 337.0189
**98**	26.21	C_42_H_35_O_28_	6′-methyl neochebulinate	303.472	292.172	987.1317	0.2	[M–H]^−^	351.0351, 205.0506, 465.0663, 169.0143, 231.0289, 337.0199, 275.0197, 307.0459, 447.0560, 617.0763, 785.0822
**107** *	28.53	C_41_H_31_O_27_	chebulinic acid	303.706	278.017	955.1061	0.8	[M–H]^−^	275.0199, 205.0506, 337.0200, 319.0094, 169.0143, 465.0668, 293.0402, 617.0781, 249.0402,785.0834, 447.0567
flavonoids									
**87**	23.29	C_21_H_19_O_11_	orientin	–	200.463	447.0926	–0.2	[M–H]^−^	297.0402, 327.0505, 285.0396, 133.0289, 311.0560, 339.0502, 357.0608
**93**	24.27	C_21_H_19_O_10_	vitexin	–	198.043	431.0991	3.0	[M–H]^−^	283.0608, 311.0568, 239.0709, 161.0243, 117.0342
**99**	26.22	C_27_H_29_O_16_	rutin	235.254	230.474	609.1460	0.7	[M–H]^−^	300.0276, 271.0236, 151.0033
**104**	27.57	C_27_H_29_O_15_	kaempferol-3-*O*-rutinoside	–	228.280	593.1503	–0.5	[M–H]^−^	285.0391, 255.0289, 229.0513, 187.0395
triterpenoids									
**102**	26.92	C_37_H_59_O_13_	MN-34	271.880	259.737	711.3989	4.6	[M+HCOOH–H]^–^	503.3389, 485.3279, 441.3377, 169.0132
**103**	27.33	C_37_H_59_O_13_	MN-35	265.590	257.723	711.3979	3.2	[M+HCOOH–H]^–^	503.3397, 453.2992, 169.0147
**109**	28.76	C_37_H_59_O_13_	MN-36	259.720	255.801	711.3954	–0.3	[M+HCOOH–H]^–^	503.3372, 169.0143, 113.0226, 59.0133
**113**	30.32	C_30_H_47_O_6_	arjungenin or terminolic acid	233.955	225.260	503.3383	2.0	[M–H]^−^	409.3106, 453.3002, 421.3100, 379.2992, 300.9970, 485.3248, 503.3364
**114**	30.49	C_30_H_47_O_6_	arjungenin or its isomer	235.768	225.454	503.3382	1.8	[M–H]^−^	503.3374, 457.3321, 409.3100, 391.2991, 73.0298
**115**	30.71	C_30_H_47_O_6_	arjungenin or its isomer	232.995	224.792	503.3383	2.0	[M–H]^−^	503.3370, 485.3272,453.3001, 409.3108, 391.3014
**117**	31.39	C_30_H_47_O_6_	arjungenin or its isomer	238.133	227.331	503.3388	3.0	[M–H]^−^	503.3378, 457.3336, 407.2948, 337.2841
**119**	32.10	C_30_H_47_O_5_	MN-37	234.349	222.987	487.3433	2.1	[M–H]^−^	487.3423, 299.9897, 410.3133
**120**	32.20	C_30_H_47_O_5_	MN-38	238.780	222.913	487.3436	2.7	[M–H]^−^	487.3417, 393.3158, 423.3254, 300.996, 467.3138, 441.3362

* The constituents identified by comparisons with the standard compounds; MNs represent those constituents identified based on analysis of the GNPS-molecular networking.

## Data Availability

The data presented in this study are available within the article and Appendix A.

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
