# Peer review of "Global Profiling of the Antioxidant Constituents in Chebulae Fructus Based on an Integrative Strategy of UHPLC/IM-QTOF-MS, MS/MS Molecular Networking, and Spectrum-Effect Correlation"

_antioxidants, 2023, doi:10.3390/antiox12122093_

Round 1
Reviewer 1 Report
Comments and Suggestions for Authors
In The manuscript titled “Global profiling of the antioxidant constituents in Chebulae Fructus based on an integrative strategy of UHPLC/IM–QTOF- MS, MS/MS molecular networking, and spectrum-effect correlation” the authors use an integrative strategy of UHPLC/IM-QTOF-MS analysis, MS/MS molecular networking, in-house library search, and CCS simulation and comparison for rapid characterization of the chemical constituents in Chebulae Fructus (CF).
I think this work is very important, it has been done with scientific rigour. I therefore accept the work after major revision
My suggestion are the following:
In the introduction I would also mention other antioxidants such as: Phenol-Rich Feijoa sellowiana (Pineapple Guava) Extracts: 10.1080/14786419.2018.1543686
I suggest that the authors explain the results in a simpler way so that the work is understandable to everyone and not only to experts in this field.
Explain the possible practical use of these metabolites
Better explain the limitations of this study in the discussion
the materials and methods are well explained. The introduction is a little too long.
Minor editing of English language required
Reviewer 2 Report
Comments and Suggestions for Authors
In the article entitled “Global profiling of the antioxidant constituents in Chebulae Fructus based on an integrative strategy of UHPLC/IM–QTOF-MS, MS/MS molecular networking, and spectrum-effect correlation” interesting and innovative research was presented. The strongest point of the manuscript is both the research material coming from various regions and fully identified, as well as the analytical methods used to identify the compounds contained in the tested material. My comments mainly concern the presentation of the results obtained and the addition of analytical details. All detailed comments are presented below.
Has the new strategy used to identify and characterize compounds in plant material been used by other researchers? If so, such information should be provided in the introduction of the work. If not, such information should also be included in the introduction of the work and the innovative aspect should be marked.
Some of the samples used in the research are fruit, while some are pulp. Is there information available (at least in some samples) on how these materials were obtained (peeling method, removal of inedible parts, drying). If there is information about the processing of these fruits, it is worth adding it to the description of the research material. Since this raw material is little known in Europe, any information related to the preparation of dried samples will be very important.
The description of analytical methods requires supplementing the missing information. How, using what device and under what conditions was the concentration to dryness process carried out? (L118) Was the extraction and isolation method based on a previously described analytical method? Have preliminary studies been carried out and the selection of conditions (e.g. solutions chosen for extraction) based on previous results? How were the CF samples pulverized? What device was used? (L195).
I recommend giving g instead of rpm when describing the centrifugation process.
Were the samples not filtered before injection into UPLC? If so, please provide the type of material from which the syringe filter was made.
The title of table number 1 is missing in the text of the manuscript.
Some of the drawings included in the article are so illegible that it is difficult to draw any conclusions from them. I recommend that drawings, e.g. 4A and C, be transferred to additional data. They are so illegible that they do not need to be in the main text of the manuscript.
The summary lacks an indication of the future direction of research. What direction should future research on the composition or antioxidant properties of this plant take? Should we move towards determining antioxidant activity using other analytical methods, should we determine the bioavailability of new identified compounds, or should we extend the research to raw materials from other regions and develop a method to determine authenticity? It is worth pointing out this direction of research.
Round 2
Reviewer 1 Report
Comments and Suggestions for Authors
Accept in present form
Comments on the Quality of English LanguageMinor editing of English language required